# PeerJ

# Sampling designs matching species biology produce accurate and affordable abundance indices

Grant Harris[1], Sean Farley[2], Gareth J. Russell[3], Matthew J. Butler[1] and Jeff Selinger[4]

[1] United States Fish and Wildlife Service, Albuquerque, NM, USA
[2] Alaska Department of Fish and Game, Anchorage, AK, USA
[3] Department of Biological Sciences, New Jersey Institute of Technology, Newark, NJ, USA
[4] Alaska Department of Fish and Game, Soldotna, AK, USA

## ABSTRACT

Wildlife biologists often use grid-based designs to sample animals and generate abundance estimates. Although sampling in grids is theoretically sound, in application, the method can be logistically difficult and expensive when sampling elusive species inhabiting extensive areas. These factors make it challenging to sample animals and meet the statistical assumption of all individuals having an equal probability of capture. Violating this assumption biases results. Does an alternative exist? Perhaps by sampling only where resources attract animals (*i.e.*, targeted sampling), it would provide accurate abundance estimates more efficiently and affordably. However, biases from this approach would also arise if individuals have an unequal probability of capture, especially if some failed to visit the sampling area. Since most biological programs are resource limited, and acquiring abundance data drives many conservation and management applications, it becomes imperative to identify economical and informative sampling designs. Therefore, we evaluated abundance estimates generated from grid and targeted sampling designs using simulations based on geographic positioning system (GPS) data from 42 Alaskan brown bears (*Ursus arctos*). Migratory salmon drew brown bears from the wider landscape, concentrating them at anadromous streams. This provided a scenario for testing the targeted approach. Grid and targeted sampling varied by trap amount, location (traps placed randomly, systematically or by expert opinion), and traps stationary or moved between capture sessions. We began by identifying when to sample, and if bears had equal probability of capture. We compared abundance estimates against seven criteria: bias, precision, accuracy, effort, plus encounter rates, and probabilities of capture and recapture. One grid (49 km$^2$ cells) and one targeted configuration provided the most accurate results. Both placed traps by expert opinion and moved traps between capture sessions, which raised capture probabilities. The grid design was least biased (−10.5%), but imprecise (CV 21.2%), and used most effort (16,100 trap-nights). The targeted configuration was more biased (−17.3%), but most precise (CV 12.3%), with least effort (7,000 trap-nights). Targeted sampling generated encounter rates four times higher, and capture and recapture probabilities 11% and 60% higher than grid sampling, in a sampling frame 88% smaller. Bears had unequal probability of capture with both sampling designs, partly because some bears never had traps available to sample them. Hence, grid and targeted sampling

Corresponding author
Grant Harris, grant_harris@fws.gov

generated abundance indices, not estimates. Overall, targeted sampling provided the most accurate and affordable design to index abundance. Targeted sampling may offer an alternative method to index the abundance of other species inhabiting expansive and inaccessible landscapes elsewhere, provided their attraction to resource concentrations.

## INTRODUCTION

Wildlife biologists often rely on photographs or DNA in a capture-mark-recapture (CMR) framework to estimate the abundances of sparsely distributed animals inhabiting expansive, heavily vegetated and inaccessible terrain (*Karanth & Nichols, 1998*; *Boulanger et al., 2002*; *Harris et al., 2010*; *Kindberg et al., 2011*). At issue is how to sample effectively and economically under such conditions. In particular, are there alternative sampling designs to the conventional grid?

Grid-based designs operate by enveloping study sites with a uniform cell size and sampling within each cell (e.g., *Woods et al., 1999*; *Poole, Mowat & Fear, 2001*; *Boulanger et al., 2002*; *Williams, Nichols & Conroy, 2002*; *Boulanger, Stenhouse & Munro, 2004*; *Kendall et al., 2009*). Although the approach is popular, grid sampling harbors logistical challenges, and when improperly used, methodological flaws (see below). An alternative sampling design could sample only at biologically important locations that attract the target species (hereafter "targeted sampling"; *Karanth & Nichols, 1998*; *Sawaya et al., 2012*). Obvious resistance to this approach stems from the biases introduced if some individuals in the population failed to visit the sampling area.

While abundance estimation forms a cornerstone of wildlife biology, the size of many study sites, challenging terrain and elusiveness of the target species makes acquiring abundance data expensive. Yet most biological programs receive little funding, requiring professionals to pursue economical methods to estimate abundance, while maintaining informative and scientifically defensible designs. Given this reality, we evaluated the targeted sampling design by comparing it to conventional grid sampling. If targeted sampling generates abundance estimates affordably and precisely, with low bias and effort, then it could offer an informative and economical alternative to estimating abundances of sparsely distributed animals inhabiting heavily vegetated or topographically complex environments, provided the species attraction to resource concentrations. Globally, many species having these characteristics are of high conservation concern (e.g., *Fuller, 1995*; *Weber & Rabinowitz, 1996*; *Shackleton, 1997*). Often, abundance data for such species are deficient, unable to inform population status, threat assessments, or provide direction for properly conserving and managing populations of these animals.

## Grid and targeted sampling

Grid sampling aims to ensure that all individuals within the population have the same probability of capture (*Nichols & Karanth, 2002*; *Williams, Nichols & Conroy, 2002*). While the theory is sound, implementation has at least 3 disadvantages. First, to ensure that all individuals encounter traps, users often scale grid cells to match the average home range size of the target species (*Boulanger, Stenhouse & Munro, 2004*; *Boulanger et al., 2006*). Problems arise when biologists lack home range information for their population under study, home range sizes vary between surveys, or home range sizes vary widely within a population. When home range data are lacking, then practitioners can rely on home range sizes from the same or related species elsewhere, hoping that the cell size used is appropriate (*Boulanger et al., 2006*; *Sawaya et al., 2012*). If animals change their behavior in response to concentrated or dispersed resources, making their home ranges shrink or expand, then home range sizes will vary between years (i.e., surveys). This makes it challenging for a defined grid size to always be appropriate. If home range sizes have much variation in the population, then a mean value has little biological relevance (*McNab, 1963*; *Powell, 2000*; *Boulanger et al., 2006*). By relying on the home range mean, a segment of the population will have home range sizes below the cell size, so traps may not be available to capture them. In each case, using home range data to guide the size of the grid cell may result in the grid being too large, causing an unequal probability of capture. This raises capture heterogeneity, thereby increasing error in the CMR approach (*Williams, Nichols & Conroy, 2002*; *Boulanger, Stenhouse & Munro, 2004*). Since many studies sampling with grid configurations do not biologically justify the grid resolution chosen, they lack information to evaluate this heterogeneity violation (e.g., *Woods et al., 1999*; *Boulanger, Stenhouse & Munro, 2004*; *Mowat et al., 2005*; *Boulanger et al., 2008a*; *Kendall et al., 2008*). In any event, for grid sampling, the solution relies on using small cell sizes to ensure that all individuals have a chance of being captured.

Of course, this begs the question of how small cell sizes must be. Although we discuss this point later, it invokes the second issue. Many study sites are large (10,000s of $km^2$) and include areas difficult to access, due to vegetation types, terrain, or insecure political situations. Sampling in these areas is logistically difficult or unsafe, and when using small cell sizes, the projects can become prohibitively expensive (*Kendall et al., 2009*; *Harris et al., 2010*; *Sawaya et al., 2012*). A third issue with grid sampling is the same trapping effort covers an entire study area that varies in species densities. This seems inefficient. Overall, these problems risk generating ineffective and expensive sampling designs, which can discourage project implementation (*Woods et al., 1999*; *Boulanger et al., 2002*; *Kendall et al., 2009*).

Resources influence a species' distribution by attracting animals from the wider landscape into relatively smaller areas. Therefore, places that concentrate important biological resources often contain a high density of the target species (*MacArthur, 1972*; *Karanth, Kumar & Nichols, 2002*). Hence, targeted sampling focuses effort in high use areas (places that attract the target animal), which may reduce the amount of area to sample, number of traps, and sampling logistics (e.g., scent marking stations, water holes,

trails; *Karanth & Nichols, 1998*; *Sawaya et al., 2012*). Each of these factors lowers project costs. Because the sampling design matches species biology and more animals are sampled in less area, it should increase capture probabilities (*Karanth, Kumar & Nichols, 2002*). Further, targeted sampling may facilitate sampling insecure study sites, if the resource concentrations – and hence high densities of the target species – occur in safer areas where they can be sampled.

The estimator would fail if numerous individuals in the population did not frequent the sampling area, or if the proportion that visited changed over time. Then, traps in a targeted configuration would not be available to all individuals equally, in amounts that could vary between surveys, causing bias. This would render the sampling assumptions unreasonable, and the final abundance estimates unreliable (*Karanth & Nichols, 1998*; *Williams, Nichols & Conroy, 2002*). Further, changes in abundances between years would reflect behavioral responses of the species, or an artifact of sampling design, and not actual variation of the species abundances. The sampling techniques would not generate estimates, but indices. Of course, these are the same issues encountered when sampling animals with grids, when home range sizes vary between surveys, or when cells are too large, so traps fall outside of some animals' home ranges rendering them unavailable for capture.

Our goal is to identify sampling designs that provide accurate, affordable and defensible abundance estimates. Therefore, we compared abundance estimates generated from grid and targeted sampling designs against a known population size. Our evaluations relied on geographic positioning system (GPS) location data from female brown bears (*Ursus arctos*) inhabiting the Kenai Peninsula of south-central Alaska, USA. Here, migratory salmon drew brown bears from the wider landscape, concentrating them at anadromous streams (i.e., streams containing anadromous fish). This provided a scenario for testing the targeted sampling approach (*Miller et al., 1997*; *Hilderbrand et al., 1999*; *Boulanger, Himmer & Swan, 2004*; *Mowat et al., 2005*). Because comparisons between designs would be nearly impossible to accomplish in the field, we simulated all sampling in a geographical information system (GIS).

We began by determining the optimal dates to sample. Next, we evaluated if all bears were available for capture during that period. For the targeted scenario, bears must be near streams with anadromous fish. For grids, traps must occur within each bear's home range. Therefore, we compared brown bear home range sizes to grid cells of varying extent, to inform the cell sizes used for sampling. For grid and targeted sampling, we sampled during the optimal periods, and evaluated each design against seven criteria: bias, precision and accuracy in the population estimates, effort required to obtain estimates (trap-nights) and encounter rates. We also evaluated capture and recapture probabilities, since the best way to reduce capture heterogeneity is by maximizing them (*Lukacs, 2009*).

When biologists estimate the abundance of animal populations, their sampling considerations include logistics: the number of traps to deploy, how to place traps, and whether to move traps between capture sessions. Their constraints are time and costs. Therefore, designs with fewer traps and systematic placement are preferable, provided they produce unbiased abundance estimates at the required precision. Similarly, since moving

traps between capture sessions adds considerable expense, it makes sense to move traps only if the improvements in accuracy outweigh those costs. Hence, our sampling designs varied by trap amount, location (i.e., traps set randomly, systematically or located by expert opinion), and traps stationary or moved between capture sessions.

## METHODS

### Study area

We studied brown bears on the Kenai Peninsula, of south-central Alaska, USA (23,310 km$^2$). The western third of the Peninsula consists of thickly forested plains (approximately sea level), while ice fields and precipitous mountains dominate the remaining landscape. Mountains peak at ≈ 2,000 m, and we never sampled areas covered by water or ice (Fig. 1). During summer, schools of anadromous salmon (five species) leave the sea and spawn in the Kenai's freshwater tributaries. These migrating salmon form a resource pulse for brown bears and other wildlife (*Hilderbrand et al., 1999*).

### GPS telemetry

Data representing bear movements came from GPS telemetry of female brown bears acquired on the Kenai Peninsula. During 1995–2003, the Alaska Department of Fish and Game captured female brown bears throughout the entire Peninsula. Captures occurred in mid-Spring, before deciduous trees leafed and salmon entered streams. Bear captures were geographically dispersed (excluding glaciers) and each bear received a GPS collar. Only females were collared, since males and cubs experience rapid weight gain throughout a season, making the collars either too loose or tight, thereby increasing the risk of injuring (or killing) the bear. All field and capture methods were approved by Alaska Department of Fish and Game, Animal Care and Use Committee, Assurance No. 06-03. GPS collars recorded geographical location, and associated date and time, at intervals ranging from 15 min to 13 h. We only used data with 3-D accuracy, and resampled these GPS data to 13 h. Resampling kept the first GPS location for each bear, and each 13-h fix thereafter. Data for some bears spanned multiple years, generating 51 bear/year combinations. Most data covered 1 June through 30 September of a given year (period of most bear activity, after winter denning).

### Trapping period

CMR studies often use 4 to 5 sampling sessions (*Karanth & Nichols, 1998*; *Boulanger et al., 2008a*; *Boulanger et al., 2008b*; *Sawaya et al., 2012*). We followed this practice, and simulated 5 sessions, each lasting 10 days (50 total days). We determined when to sample by identifying the period between 1 June and 30 September that bears would most likely encounter traps.

We examined 3 approaches to identify the sample period for the grid-based designs. First, we hypothesized that bears leaving dens would explore the landscape and encounter traps. For bears in Alaska, this period began 1 June and ended 20 July. Next, we presumed that greater movements would cause more trap encounters. Therefore, we found the contiguous window of 50 days, when the average movement rate for brown bears between

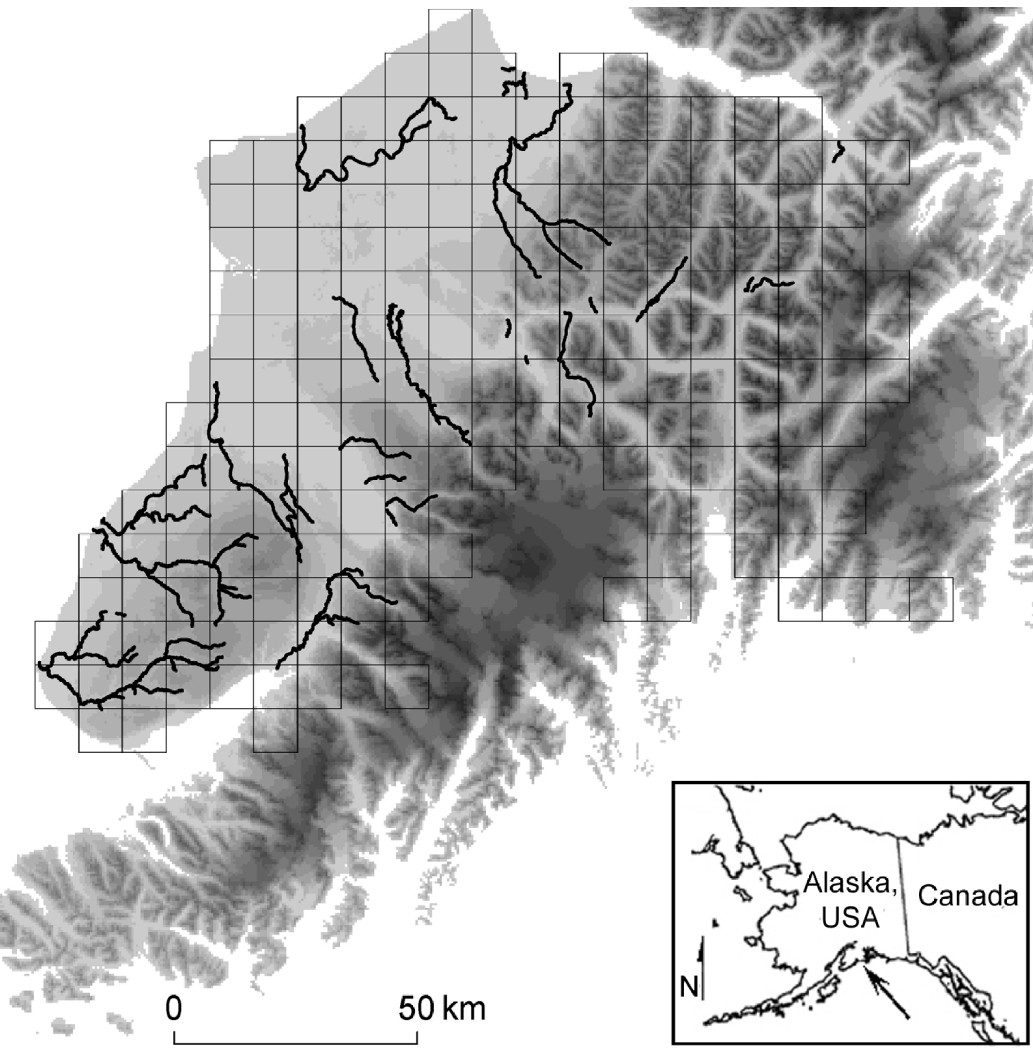

**Figure 1 Study Location.** We simulated and compared capture-mark-recapture using grid-based and targeted sampling designs for brown bears (*Ursus acrtos*) inhabiting the Kenai Peninsula (center, with elevation shaded from low [light gray] to high [black]), south-central Alaska, USA (inset). The grid-based design used cells with an area of 49 km², 81 km² (pictured in gray), and 121 km². The targeted design sampled places where important biological resources concentrated the target species (i.e., bear's attraction to anadromous streams; black lines).

successive GPS fixes was maximized. Lastly, we examined long-range movements by identifying the contiguous window of 50 days that maximized the average distances between the furthest-apart GPS fixes for each bear.

To calculate the movement metrics, we divided the 1 June through 30 September period into 10-day brackets. This generated 122 different intervals (day 1–10, day 2–11, etc.). We then quantified the mean and standard error of brown bear movements in each of the intervals. We retained the 5 contiguous intervals that maximized bear movements (50 contiguous days).

The targeted design hinged on salmon being an important food resource that attracts brown bears to anadromous streams (*Hilderbrand et al., 1999*; *Beier et al., 2005*). Therefore, we used the same 122, 10-day intervals to quantify the proportion of GPS fixes ≤500 m of an anadromous stream for each bear. We chose the 5 contiguous 10-day intervals that maximized this proportion. Targeted simulations occurred during those dates.

## Availability for capture

We quantified the sizes of brown bear home ranges, to inform the selection of appropriate cell sizes for the grid-sampling frame. The objective was to ensure that all bears were available for capture. Our method examined the relationships between grid-cell size (km), and the proportion of bears with home range sizes below that cell's area. We also calculated the number of traps within each bear's home range, to determine if it contained at least one trap. For each bear, home range calculations used GPS data spanning 1 June through 20 July of a given year, to match when grid sampling occurred (see Results). To estimate home range, we used 95% fixed-kernel and least-squares cross-validation to calculate the smoothing parameter (*Hooge & Eichenlaub, 2000*).

Data represented home range sizes of different classes of bears (females without cubs, females with cubs of the year, and females with cubs over one year old). In practice, biologists often do not know in advance what classes, or numbers of individuals within those classes would be trapped, so they usually rely on the average home range size to inform the area of the sampling cell (*Boulanger, Stenhouse & Munro, 2004*; *Boulanger et al., 2006*). To test the notion of using an average home range size to inform grid cell area, we averaged the sizes of all home ranges together ($n = 42$ bears with GPS data spanning the 1 June through 20 July period). We also reported the home range size of just females with cubs ($n = 29$).

For targeted sampling, we also determined the proportion of bears available for capture. We relied on the GPS data to evaluate if each bear was within 500 m of an anadromous stream during the sampling period. We then examined the proportion of bears within this sampling frame, for every year of GPS data. The sampling frame was about 3,500 km$^2$, or 22% of the grid-sampling frame ($\approx$16,000 km$^2$).

## Sampling designs

For grids, we simulated traps in a GIS by dividing the peninsula into 3 sets of square cells, with areas of 49, 81 and 121 km$^2$ (Table 1). These cell areas were informed by the home range analysis, and similar cell sizes are often used to estimate brown and grizzly bear population sizes (e.g., *Boulanger et al., 2002*; *Beier et al., 2005*; *Mowat et al., 2005*; *Boulanger et al., 2008b*; *Kendall et al., 2008*). Traps were stationary between capture sessions and located using 3 different configurations: traps in cell centers, traps placed randomly in the cell, or traps placed by expert opinion in each cell. Expert opinion relied on our knowledge of bear behavior to locate traps (in the GIS) where bears would most likely intercept them, were the traps actually set on the ground. This included locations along streams, ravines and similar topographical features that can attract or funnel moving bears.

**Peer**J

**Table 1 Sampling configurations.** Characteristics of the sampling configurations used to simulate capture-mark-recapture of brown bears on the Kenai Peninsula, south-central Alaska, USA. Attributes included the timing of the simulation (period), position of traps (placement), whether traps were stationary or moved between capture sessions, cell area (km$^2$) or trap spacing (km), the number of traps in each of 5 capture sessions, and effort (trap-nights).

| Sampling configuration | Period | Placement | Stationary (S) or Moved (M) | Cell area or spacing | No. traps per capture session | Total effort (trap-nights) |
|---|---|---|---|---|---|---|
| Grid | 6/1–7/20 | Cell Center | S | 49, 81, 121 | 322, 195, 135 | 16,100, 9,750, 6,750 |
| Grid | 6/1–7/20 | Random | S | 49, 81, 121 | 322, 195, 135 | 16,100, 9,750, 6,750 |
| Grid[a] | 6/1–7/20 | Expert opinion | S | 49, 81, 121 | 322, 195, 135 | 16,100, 9,750, 6,750 |
| Grid[a] | 6/1–7/20 | Expert opinion | M | 49, 81, 121 | 322, 195, 135 | 16,100, 9,750, 6,750 |
| Targeted | 7/10–8/28 | Systematic | S | 17, 19 | 324, 312 | 16,200, 15,600 |
| Targeted[a] | 7/10–8/28 | Expert opinion | S | N/A | 140, 70 | 7,000, 3,500 |
| Targeted[a] | 7/10–8/28 | Expert opinion | M | N/A | 140, 70 | 7,000, 3,500 |

**Notes.**
[a] These configurations were analyzed and compared (each configuration had 30 simulations). The other configurations produced inaccurate abundances.

We simulated the targeted designs using two methods, each with effort (trap-nights) comparable to or less than the grid configurations (Table 1). First, traps were systematically spaced in the GIS, with 17 and 19 km spacing (along anadromous streams). Second, trap locations were identified in the field, by expert opinion. We located these places by ground and plane reconnaissance, based on the characteristics of shallow water or narrow reaches, since these attributes facilitate bears' ability to catch fish. Trap coordinates were input into the GIS. The traps placed by expert opinion had two configurations that varied by effort.

Previous studies suggested that moving traps between captures sessions increases capture probabilities and the number of individuals captured (*Boulanger et al., 2002*; *Boulanger, Stenhouse & Munro, 2004*; *Boulanger et al., 2006*). This should reduce bias and error in abundance estimates. We tested this advice for brown bears using grid and targeted sampling designs, with traps placed by expert opinion. For grids, we began by simulating 5 different sets of trap locations within each cell. Trap locations were identified with aerial photography of the study area, and the coordinates input into a GIS. We examined 30 scenarios of stationary traps (for each cell size). For five of these scenarios, we held each of the trap sets (sets 1–5) constant across all capture sessions. For the remaining 25 scenarios, we randomly picked a trap in each cell, and held this set constant across all capture sessions (Table 1).

When traps were moved between capture sessions, a different trap set (1 of the 5) was used once for each capture session. For example, trap set 3 in capture session 1, then trap set 4 in session 2, etc. We randomly selected 30 of the 120 possible permutations of moved traps within each cell (separately for grids of 49, 81, and 121 km$^2$; Table 1).

For targeted scenarios, we iteratively picked every 5th trap from the set of expertly located traps, to ensure a different trap was used each session (140 traps/session). A second configuration of targeted sampling halved this number of traps (70 traps/session). Permutations of stationary and moved traps followed the same procedure outlined above (Table 1).

Lastly, we generated a hybrid technique. This design had traps expertly placed along streams, and traps in a grid formation at 9 km spacing, located $\geq 15$ km from anadromous streams.

## Abundance estimation and evaluation

For each sampling configuration, we recorded a capture when a bear's GPS location occurred $\leq 500$ m of a trap. All abundance estimates were then calculated with closed capture models in the program MARK (version 6.2; *White & Burnham, 1999*). We fit 8 models under the "full closed captures with heterogeneity" data type. The first model was $M_o$, which assumed no variation in capture probability associated with individuals or occasions (only one capture probability, $p$). Model $M_t$ assumed variation in capture probability between sampling sessions, but not among individuals within a session. Model $M_h$ (heterogeneity) permits different capture probabilities for individuals, which remains the same over all sampling occasions regardless of the capture history (*Karanth & Nichols, 1998*). The fourth model, $M_b$, allows bears to demonstrate behavioral responses (e.g., trap happy or shyness), an effect which can exist in simulations (*White, 2008*). The remaining models allowed for combinations of the previous models ($M_{th}, M_{tb}, M_{bh}, M_{tbh}$).

We used the program MARK to estimate abundance and the related metrics, based on the models receiving the most parsimonious fit. We used second-order Akaike's Information Criterion ($AIC_c$) for model selection and treated models with $\Delta AIC_c \leq 2$ as competitive (*Anderson & Burnham, 2002*; *Burnham & Anderson, 2002*). We averaged the parameters of competitive models based on model weights.

We evaluated abundance estimates for each grid and targeted sampling design against 7 criteria. First, we calculated average bias (how close the population estimate was to the true abundance value) as:

$$Bias(\hat{N}) = \frac{1}{n}\sum_{i=1}^{n}(\hat{N}_i - N)$$

with $\hat{N}$ representing the mean abundance estimate, $\hat{N}_i$ the abundance estimate for simulation $i$, $N$ the true abundance and $n$ the number of simulations (30). We reported this as a relative score, by dividing bias by true abundance ($N = 42$). Next, we examined precision, the degree to which an individual estimate is repeatable or reproducible. We reported coefficient of variation (CV) as a proportional measure of error in the abundance estimates

$$CV(\hat{N}) = 100 * \frac{SD(\hat{N})}{\hat{N}}.$$

Accuracy is a combination of these metrics (*Williams, Nichols & Conroy, 2002*), calculated as:

$$RMSE(\hat{N}) = \sqrt{\frac{1}{n}\sum_{i=1}^{n}(\hat{N}_i - N)^2}$$

$RMSE(\hat{N})$ was reported relatively, by dividing by true abundance ($N = 42$). We also reported encounter rates, capture probabilities, and recapture probabilities. Encounter rates described the number of GPS fixes considered captures, divided by the total number of GPS fixes (for each bear). Capture probabilities were the number of bears caught divided by the total number of known bears. Recapture probabilities were calculated as the number of bears caught more than once, divided by the total number of bears captured. We used 2-sample $t$-tests to compare the number of bears captured and recaptured between stationary and moved traps for each scenario (*Zar, 1999*). If Levene's test for equality of variances indicated unequal variances, we used $t$-tests where equal variances were not assumed (*SPSS, 2010*).

We also calculated the amount of effort (trap-nights) required to acquire population estimates as a surrogate for cost. This value was the number of traps multiplied by the number of days sampled.

Lastly, we used program TRENDS to determine the number of survey years required to detect a given population change or trend (*Gerrodette, 1993*). Parameters were $\alpha = 0.05$, $1 - \beta$(power) $= 0.8$, $CV(\hat{N}) \propto \sqrt{\hat{N}}$, with a linear change over time. This exercise assisted with interpreting the influence of $CV(\hat{N})$ on the effectiveness of each scenario for monitoring abundance changes.

## RESULTS

### Sampling periods

We analyzed data from 42 female brown bears to evaluate the grid and targeted sampling designs. We based the grid simulations during the 1 Jun–20 Jul period, since the greatest number of brown bears encountered traps in spring. There was an average of 57.0 GPS fixes per bear (SD 11.0) during this time. The targeted scenarios occurred between 10 July through 28 August, when brown bears had most location fixes $\leq$500 m of anadromous streams. On average, there were 58.0 GPS fixes per bear (SD 20.1) during this sampling period.

### Availability for capture

For grid-based and targeted sampling to produce reliable population estimates, all individuals in the population must have traps available to capture them. To accomplish this with grids, practitioners recommend that cell area match the average home range size of the target species (*Boulanger, Stenhouse & Munro, 2004*; *Boulanger et al., 2006*; *Sawaya et al., 2012*). Therefore, we calculated home range sizes for the brown bears we sampled. The mean value was 149.7 km$^2$ (SD 149.8 km$^2$). When considering home range sizes of females with cubs, the mean home range size was 141.3 km$^2$ (SD 145.8 km$^2$). A $12 \times 12$ km cell approximates either area (Fig. 2).

Sixty-seven percent of the bears had home range sizes below this value. A cell size of 25 km$^2$ would miss <5% of bears yet the effort involved in employing such a grid is logistically untenable (30,050 trap-nights) and would be exorbitantly costly for a study site this large. We evaluated grids with edge lengths of 7, 9 and 11 km, and the following
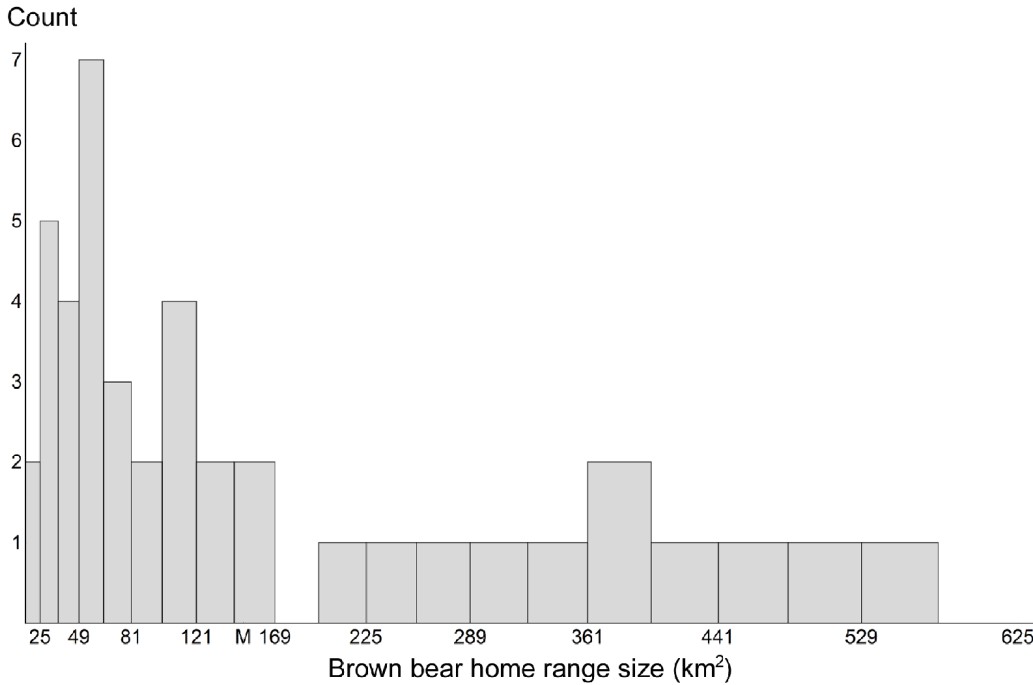

**Figure 2 Brown bear home range sizes.** Distribution of home range sizes of brown bears ($n = 42$) during 1 June through 20 July on the Kenai Peninsula, south-central Alaska, USA. Mean home range size (M) was 149.7 km$^2$ (SD = 149.8 km$^2$). Sixty-seven percent of bears had home ranges smaller than the mean.

proportions of bears had home range sizes below each of these areas, respectively: 26%, 50% and 64%.

However, traps (especially when expertly placed) may occur within the animals' home range, even if it is smaller than the cell size employed. Therefore, for each grid scenario (expertly placed traps) we determined if each bear had a trap within its home range. The average number of bears without traps in their home range for stationary traps was 4.4 (SD 1.9; 49 km$^2$), 9.5 (SD 3.2; 81 km$^2$), and 14.8 (SD 2.1; 121 km$^2$). Moving traps between capture sessions reduced these values, whereby the following numbers of bears never had a trap within its home range 1 (49 km$^2$), 1 (81 km$^2$), and 3 (121 km$^2$).

For targeted sampling, the sampling frame was the area within 500 m of an anadromous stream. All bears were located within 500 m of anadromous streams during the sampling session, except for 1997, when 2 bears were unavailable (although these bears were ≤500 m of streams after the sampling session; Table 2).

Because one to three bears were always unavailable for capture, by definition, grid-based and targeted sampling generated population indices, not estimates (hereafter termed indices). As indicated, moving traps attenuates the problem of ensuring that all bears have equal opportunity of encountering traps, but did not fix it.

## Evaluation of sampling configurations

For each sampling configuration, we evaluated abundance indices, encounter rates, capture probabilities, recapture probabilities, and the effort required to acquire them.

**Table 2 Availability for capture.** The total number of bears on the Kenai Peninsula during the sampling period for the targeted scenario, along with the number of bears present within the sampling frame ($\leq$500 m of an anadromous stream) during that period, for each year of sampling.

| Year | Total no. bears in sampling period | No. bears within sampling frame |
|------|-----------------------------------|--------------------------------|
| 1996 | 2.0 | 2.0 |
| 1997 | 14.0 | 12.0 |
| 1998 | 4.0 | 4.0 |
| 1999 | 8.0 | 8.0 |
| 2000 | 6.0 | 6.0 |
| 2003 | 5.0 | 5.0 |
| 2004 | 3.0 | 3.0 |

For grids, the configurations with traps in cell centers or randomly placed produced the lowest capture probabilities, and generated abundance indices with high bias and error. Targeted sampling with traps systematically placed also generated inaccurate abundance indices, and the hybrid configuration provided results similar to targeted sampling alone. Therefore, we did not present or consider these sampling designs further.

We focused on configurations with traps expertly placed in each cell (Table 3). Within each configuration, sampling with traps moved between capture sessions always provided more accurate population indices than sampling with stationary traps, mainly because moving traps halved bias. Moving traps always increased precision for targeted sampling, but inconsistently for grids (Table 3).

Sampling with traps moved between capture sessions using 49 km$^2$ cells and targeted sampling generated the most accurate population indices (Table 3). The 49 km$^2$ grid had least bias ($-10.5\%$) and low precision (CV 21.2%), while targeted sampling had more bias ($-17.3\%$) and most precision (CV 12.3%). Their accuracy was nearly equivalent (RMSE = 21.7% and 20.0% respectively; Table 3).

Grid-based sampling with moved traps in 81 km$^2$ cells and with stationary traps in 49 km$^2$ cells also generated similar accuracy (RMSE = 26.5% and 25.6%). The 81 km$^2$ configuration had less bias ($-13.0$), and more error (CV 26.6%), while stationary traps in the 49 km$^2$ grid had greater bias ($-21.5$) and less error (CV 17.8%). Sampling in the 121 km$^2$ cells or with half the amount of targeted traps (stationary) generated the most inaccurate indices, due to combinations of high bias and imprecision (Table 3).

Usually, the techniques used to estimate or index abundance should be robust enough to identify a specified level of change, be it the time necessary to quantify a given population trend, or the time required to measure a set difference between individual surveys. For example, on the Kenai Peninsula, this brown bear population has a growth rate ($\lambda$) of 1.04 (S Farley, unpublished data). At best, it would take $\geq$16 years to detect this trend with grid-based sampling (49 km$^2$ grids with stationary traps). The targeted configuration required 12 years (moved traps; Table 4). A doubling of population size is detectable in 19 or 28 years using a grid of 49 km$^2$ with traps stationary or moved, respectively. Targeted

**Table 3 Comparison of sampling configurations.** Results describing the bias, precision and accuracy of abundance estimates generated from the grid and targeted sampling configurations, with trap placement stationary or moved between capture sessions ($N = 42$ for each configuration).

| Configuration | Placement | $n$[a] | $\hat{N}$[b] | SD($\hat{N}$)[c] | %Bias[d] | %RMSE[e] | CV($\hat{N}$)[f] |
|---|---|---|---|---|---|---|---|
| Grid 7 km | Stationary | 30 | 32.98 | 5.88 | −21.48 | 25.63 | 17.82 |
| | Moved | 30 | 37.59 | 7.98 | −10.51 | 21.70 | 21.22 |
| Grid 9 km | Stationary | 30 | 29.96 | 10.57 | −28.67 | 38.16 | 35.30 |
| | Moved | 30 | 36.56 | 9.72 | −12.95 | 26.51 | 26.58 |
| Grid 11 km | Stationary | 30 | 19.49 | 9.76 | −53.60 | 58.42 | 50.10 |
| | Moved | 30 | 32.21 | 17.45 | −23.31 | 47.63 | 54.16 |
| Targeted | Stationary | 30 | 28.23 | 4.22 | −32.77 | 34.28 | 14.94 |
| | Moved | 30 | 34.75 | 4.26 | −17.27 | 20.02 | 12.26 |
| Targeted(1/2) | Stationary | 30 | 20.36 | 5.00 | −51.51 | 52.87 | 24.56 |
| | Moved | 30 | 27.42 | 4.32 | −34.71 | 36.20 | 15.75 |

**Notes.**

[a] Sample size.

[b] Mean estimated abundance.

[c] Standard deviation of mean estimated abundance.

[d] Percent relative bias.

[e] Percent relative root mean squared error (accuracy).

[f] Coefficient of variation (precision).

**Table 4 Time required to measure population change.** Time (years) required to measure a lambda of 1.04, and a doubling or halving of overall population size, from 5 different sampling configurations with traps stationary (S) or moved (M) between capture sessions. Calculations relied on CVs reported in Table 3.

| Sampling configuration | $T_\lambda$[a] | $T_d$[b] | $T_h$[c] |
|---|---|---|---|
| Grid 7 km (S) | 16 | 19 | 6 |
| Grid 7 km (M) | 19 | 28 | 7 |
| Grid 9 km (S) | 34 | 61 | 17 |
| Grid 9 km (M) | 23 | 43 | 11 |
| Grid 11 km (S) | 48 | 82 | 36 |
| Grid 11 km (M) | 54 | 106 | 41 |
| Targeted (S) | 14 | 13 | 4 |
| Targeted (M) | 12 | 10 | 4 |
| Targeted(1/2) (S) | 21 | 36 | 9 |
| Targeted(1/2) (M) | 15 | 16 | 5 |

**Notes.**

[a] Number of years required to detect a yearly growth rate ($\lambda$) of 1.04.

[b] Number of years required to detect a doubling of the overall population size.

[c] Number of years required to detect a halving of the overall population size.

**Table 5 Encounter rates.** Mean and standard deviation of encounter rates (number of GPS locations considered captures/total number of GPS locations) for all bears in the sampling period, for 5 grid and targeted sampling configurations, with traps stationary or moved between capture sessions ($N = 42$ bears, with 30 simulations for each configuration).

| Configuration | Stationary | | Moved | |
|---|---|---|---|---|
| | $\bar{X}^a$ | SD | $\bar{X}^a$ | SD |
| 49 km$^2$ grid | 0.03 | 0.01 | 0.03 | 0.00 |
| 81 km$^2$ grid | 0.02 | 0.01 | 0.02 | 0.00 |
| 121 km$^2$ grid | 0.01 | 0.01 | 0.01 | 0.00 |
| Targeted | 0.12 | 0.03 | 0.12 | 0.01 |
| Targeted(1/2) | 0.06 | 0.02 | 0.06 | 0.01 |

**Notes.**
[a] Encounter rates.

**Table 6 Comparison of capture probabilities.** Mean and standard deviation of capture probabilities for 5 sampling configurations. We used a $t$-test to compare capture probabilities.

| Configuration | Stationary traps | | Moved traps | | Levene's test | | $t$-test | | |
|---|---|---|---|---|---|---|---|---|---|
| | $\bar{p}^a$ | SD($\bar{p}$) | $\bar{p}^a$ | SD($\bar{p}$) | F | $p$-value | t | df | $p$-value |
| 49 km$^2$ grid | 0.60 | 0.08 | 0.66 | 0.07 | 2.82 | 0.099 | $-2.92$ | 58.0 | 0.005 |
| 81 km$^2$ grid | 0.44 | 0.09 | 0.52 | 0.06 | 2.56 | 0.115 | $-3.95$ | 58.0 | 0.000 |
| 121 km$^2$ grid | 0.32 | 0.07 | 0.38 | 0.07 | 0.20 | 0.656 | $-3.31$ | 58.0 | 0.002 |
| Targeted | 0.61 | 0.07 | 0.73 | 0.05 | 2.74 | 0.103 | $-8.16$ | 58.0 | 0.000 |
| Targeted(1/2) | 0.43 | 0.10 | 0.57 | 0.05 | 10.61 | 0.002 | $-7.01$ | 44.8 | 0.000 |

**Notes.**
[a] Capture probability.

sampling required 10 years (moved traps). Were a population to halve in size, the same grid scenarios would require 6 and 7 years to detect it, while targeted sampling would take 4 years (Table 4).

## Encounter rates, capture probabilities, recapture probabilities and effort

Encounter rates within each grid and targeted configuration remained identical between traps stationary or moved (moving traps reduced variability). Encounter rates for the targeted scenario (moved traps) was 4 times greater than the best grid scenario (49 km$^2$; Table 5).

Within each configuration, moving traps always resulted in higher capture probabilities than stationary traps. The increases were between 10 and 33% (Table 6). Targeted sampling generated the highest capture probabilities. With moved traps, capture probabilities were 11% higher than sampling within the 49 km$^2$ grid ($t = -5.1$, $df = 52.2$, $p$-value $< 0.001$).

Within a given configuration, moving traps always reduced recapture probability. The result was significant for every configuration except when sampling within 49 km$^2$ grids (Table 7). The recapture probabilities declined with increasing cell size (49 km$^2$ and

**Table 7 Comparison of recapture probabilities.** Mean and standard deviation of recapture probabilities for 5 sampling configurations. We used a *t*-test to compare recapture probabilities.

| Configuration | Stationary | | Moved | | Levene's test | | *t*-test | | |
|---|---|---|---|---|---|---|---|---|---|
| | $\bar{c}$[a] | SD($\bar{c}$) | $\bar{c}$[a] | SD($\bar{c}$) | F | *p*-value | t | df | *p*-value |
| 49 km² grid | 0.47 | 0.13 | 0.43 | 0.08 | 4.05 | 0.049 | 1.19 | 49.2 | 0.241 |
| 81 km² grid | 0.42 | 0.12 | 0.34 | 0.10 | 0.11 | 0.747 | 2.79 | 58.0 | 0.007 |
| 121 km² grid | 0.48 | 0.10 | 0.29 | 0.10 | 0.01 | 0.934 | 7.36 | 58.0 | 0.000 |
| Targeted | 0.75 | 0.08 | 0.69 | 0.06 | 1.48 | 0.230 | 3.73 | 58.0 | 0.000 |
| Targeted(1/2) | 0.70 | 0.09 | 0.53 | 0.08 | 0.17 | 0.678 | 7.52 | 58.0 | 0.000 |

**Notes.**

[a] Recapture probability.

81 km² grids), or with fewer targeted traps. For grids, recapture probabilities were greater for 121 km² cells with stationary traps, because the scenario caught fewer bears more often (Table 7). We found that targeted sampling with moved traps had higher recapture probabilities than sampling in 49 km² grids ($t = -13.2$, $df = 58$, *p*-value $< 0.001$).

Lastly, effort (measured by trap-nights) was highest for grid sampling with 49 and 81 km² cells, as these scenarios expended 16,100 and 9,750 trap-nights each. Effort for the targeted configuration was 57% less than the 49 km² grids and 28% less than 81 km² grids (7,000 trap-nights; Table 1).

# DISCUSSION

Establishing defensible and economical sampling methods to obtain species abundances will inform conservation measures and management actions for many species inhabiting large, remote or complicated landscapes. Therefore, we used a CMR framework and evaluated sampling designs which varied by configuration, effort (trap amount) and complexity (stationary or moved). The goal was to identify informative and affordable sampling designs that biologists could use for estimating or indexing abundance.

For grid-based sampling, it is often recommended that cell size corresponds to the average area of an animal's home range, or smaller, to reduce capture heterogeneity (*Boulanger, Stenhouse & Munro, 2004*; *Boulanger et al., 2006*; *Sawaya et al., 2012*). However, animal home ranges usually have much variability, which can change within and across seasons based on resource pulses and their distribution (*McNab, 1963*; *Boulanger et al., 2006*; *Nielsen et al., 2010*). Therefore, a value describing the average home range size may not be biologically meaningful, making it risky to justify grid resolution on this metric. In our case, cell sizes of 49, 81 and 121 km² seemed appropriate since female brown bears had home range sizes averaging 149.7 km². Yet these cell sizes were still too large for grid sampling to provide population estimates, since some bears were not available for capture.

When grid configurations are used, justification for the cell size should not rely on the average home range size of the target species. Instead, justification falls on making cell size as small as project costs allow. However, to make studies affordable, the temptation is for fewer traps to be employed, and therefore, increased cell size. Yet we found that as the

grid cells became larger, fewer traps caught fewer bears, which lowered encounter rates, capture and recapture probabilities (Tables 5–7). This produces results with more bias and imprecision (Table 3). Given this information, how does one know when cell sizes are too large?

Unless preliminary work provides information for understanding a species' use of the landscape, then the cell size chosen will be a speculation. As a result, the abundance estimate generated may indicate an estimate or an index, or be completely uninformative.

Previously, many projects that estimated brown bear abundance operated in places spanning $\approx$2,000–9,000 km$^2$ (e.g., *Poole, Mowat & Fear, 2001*; *Boulanger et al., 2002*; *Mowat et al., 2005*; *Boulanger et al., 2006*; *Kendall et al., 2008*). Our site spanned 16,000 km$^2$. For areas of this size, small grids grow prohibitively expensive to use. In our case, cell size should be <49 km$^2$, requiring >322 traps and >16,100 trap-nights (>0.02 traps/km$^2$). Checking these traps in remote locations usually requires the use of helicopters, which often exceed $700.00 per flight hour. Project costs can easily approach $1,000,000.00 (US) in such large areas, when sampling in grids at or below this resolution. In any event, grid sampling places many traps in low density, to sample low densities of animals across an expansive area. It seems costly and inefficient.

For the targeted design, species biology informed the sampling frame, as anadromous streams attracted bears from the wider landscape. The technique was more efficient, by operating 140 traps in a 3,500 km$^2$ area (0.04 traps/km$^2$). It sampled higher densities of animals in a smaller area with less traps, generating the highest capture and recapture probabilities (Tables 6 and 7). Its accuracy was equivalent to the grid index (49 km$^2$; Table 3).

For brown bears in Alaska, the targeted configuration is easier and cheaper to operate in the field for at least three reasons. First, it required half the effort of the 49 km$^2$ grids. Second, since traps occur along streams, they are more accessible by walking or rafting. Third, the lure (fish) is provided naturally, reducing time, expense and costs for trap set-up, and concerns that the target species may lose interest in the trap over time (*Boulanger, Himmer & Swan, 2004*; *Boulanger et al., 2008a*).

The conventional assumption is that moving traps raises capture probabilities, by increasing the proportion of bears captured, and therefore reducing bias (*Boulanger et al., 2002*; *Boulanger, Stenhouse & Munro, 2004*; *Boulanger et al., 2006*). Our simulation confirmed this assumption for grid and targeted sampling, as moving traps always increased capture probabilities. Moving traps between capture sessions raised capture probabilities, because moving traps to new areas increased their availability to more bears that would otherwise not have the opportunity to encounter them. However, moving traps also caused a decline in recapture probabilities. This makes sense, as moving traps takes them from a location where they already sampled bears, and places them someplace new. Then, it becomes more difficult to recapture those same bears already caught, as the trap is more likely to be moved to a spot where they do not occur. The 49 km$^2$ grid had the most traps, which may explain why the effects of relocating traps was not as marked.

For indexing species abundances, the most accurate results occurred with traps moved between capture sessions, and traps placed by expert opinion. Moving traps reduced capture heterogeneity by increasing the number of animals in the sampling frame with a nonzero probability of encountering a trap (*Williams, Nichols & Conroy, 2002*; *Lukacs, 2009*). Unfortunately, moving traps did not ensure trap availability to all bears, and therefore did not completely alleviate the sampling bias caused by using too large a cell, or sampling at anadromous streams. All individuals in the population must visit the sampling area to generate valid estimates of abundance. Hence, the final abundances generated from both sampling designs represented population indices. Providing population estimates would require smaller grid cells, or more traps in the targeted design.

## Sampling synopsis

The most accurate configurations, targeted and grid-based sampling with 49 km$^2$ cells, had tradeoffs in bias, precision and effort (costs). Targeted sampling was more biased and precise while the 49 km$^2$ grid was less biased and less precise. Because abundances gained from grid sampling tend to be imprecise, some projects incorporate ancillary information (i.e., covariates) to increase precision, or data from captures gained elsewhere (*Boulanger et al., 2008a*; *Kendall et al., 2009*; *Sawaya et al., 2012*). Many applications to index abundance for bears and other species will not have such ancillary data available, and therefore we did not include covariates in our analyses.

Often, management applications center on evaluating trends in abundances over time. Because this exercise requires high precision, most wildlife surveys aim for a CV $\leq$ 20% (*Boulanger et al., 2002*; *Williams, Nichols & Conroy, 2002*). Otherwise, it takes unacceptably long to measure population changes and trends in time for managers to act. The most accurate targeted scenario met the criteria, by generating a CV of 12%, while the most accurate 49 km$^2$ grid did not.

When using the program MARK, the most competitive models for grids (moved traps) typically contained $M_o$ models (80% 49 km$^2$, 86% 81 km$^2$), and rarely $M_h$ based models. This outcome is indicative of grid sampling reducing heterogeneity between capture sessions. However, $M_o$ based models are sensitive to violations in the underlying model assumption of homogenous capture probabilities (*Otis et al., 1978*). For the targeted approaches, we anticipated variation in capture probability among individuals, and found that all targeted configurations (moved traps) used $M_h$ based models. The $M_h$ models are robust to violations in underlying model assumptions, providing greater confidence in those abundance estimates (*Burnham & Overton, 1978*; *Otis et al., 1978*; *Karanth, Kumar & Nichols, 2002*).

Given this synopsis, targeted sampling provides results with accuracy comparable to grid sampling, only more affordably. Hence, targeted sampling opens opportunities for indexing abundances of animal populations for many conservation and management projects, provided that the species are attracted to resource concentrations. Often, these projects are cost limited, large in geographical size, focus on rare and sparsely distributed species, and cover locations with challenging access. The abundance indices gained would

contribute toward understanding population changes between periods and measuring the trajectory of populations over time, for many species whose current information is deficient.

## Encounter distance and use of lure

The effective distance that a bear would be attracted to a scent or visual lure is unknown. The distance depends on the type of lure used, amount of elapsed time since lure deployment (more time dilutes lure effectiveness), topography, weather (e.g., aridity, precipitation), wind speed and direction, bear gender, experience and behavior (i.e., waning interest in lure; *Boulanger, Stenhouse & Munro, 2004*; *Boulanger et al., 2008a*; *Sawaya et al., 2012*). Potentially, future studies aimed at quantifying this distance could leverage off projects that estimate population sizes of lions (*Panthera leo*) and hyenas (*Crocuta crocuta*) using call playbacks as lure (*Mills, Juritz & Zucchini, 2001*; *Kiffner et al., 2008*).

Our simulation considered distances between a bears GPS location and hypothetical trap ≤500 m as encounters. The actual distance used is essentially irrelevant for the simulation. What matters is bear density in a given area. Areas with high bear density will get more captures than a place with low density, regardless of the distance between bear and trap. In our example, targeted sampling occurred in a confined area, defined as places within 500 m of anadaramous streams. Since all bears visit this relatively small area – about 88% smaller than the grid area – then traps placed here catch more bears more often. The wider landscape has lower density of bears, so traps placed there catch fewer bears less often. Preliminary work with a 100 m encounter distance generated similar outcomes to those presented here, only with less encounters and captures (G Harris, unpublished data). Seemingly, by considering larger encounter distances, it would generate more encounters and captures.

Because our effort is a simulation, it did not account for baiting traps, a common practice to attract bears when field sampling (e.g., *Woods et al., 1999*; *Boulanger et al., 2002*; *Sawaya et al., 2012*). Were a field study to occur, with traps baited, then we would expect encounter probabilities to increase, for grid and targeted sampling alike. Both sampling designs can have lure associated with them (*Sawaya et al., 2012*). Therefore, in the simulation, since data are handled identically, there is no more attraction for a bear to visit a targeted snare any more than a grid snare. Additionally, the behavioral responses measured in the modeling can reflect survey configuration (*White, 2008*).

As above, what differs is bear density in an area. The targeted sampling frame matched species biology by relying on animals' attraction to resource concentrations (*Karanth, Nichols & Kumar, 2004*). This outcome enabled targeted sampling to provide accurate abundance indices. The targeted approach is not new, and has been used elsewhere to provide abundance data with informative results, be it rub trees to collect bear hair (e.g., *Kendall et al., 2008*; *Kendall et al., 2009*; *Sawaya et al., 2012*) or to gain photographs of tigers along trails (*Karanth & Nichols, 1998*).

## Spatial capture-recapture models

We relied on closed capture models in the program MARK (*White & Burnham, 1999*) to quantify abundances, instead of other approaches such as spatially explicit capture-recapture models (SECR; *Efford, Dawson & Borchers, 2009*; *Efford, 2011*; *Royle et al., 2011*). While the SECR approach can apply to grid sampling, and we applaud the innovation, there are five reasons why we employed MARK. First, MARK is a proven method and appropriate for estimating animal abundances. Second, this project is a simulation, with all bears completely within the study frame, and the population closed (based on the GPS location data). This eliminated any ambiguity over the effective sampling area, which the SECR models rigorously address for the purposes of density estimation. Third, when grid cells exceed home range size – as encountered herein – then not all individuals are available for capture. SECR models do not solve this bias. Fourth, the SECR models rely on information describing where and when a given animal is trapped (session number) to identify each individual animals "activity center". The probability of detection for each trap is modeled as a function of the distance from the activity center to traps in the array. If the method relies on animal movements to estimate home range centers (*Royle et al., 2009*), then SECR requires multiple captures. Yet, over one-third of these bears had only one capture (each, for 49 km$^2$ and 81 km$^2$ grid cells with moved traps). This renders the SECR technique less effective. Fifth, some SECR models assume that activity centers are uniform and symmetric (*Efford, Dawson & Borchers, 2009*; *Efford, 2011*; *Royle et al., 2011*). Unfortunately, this assumption is untenable for brown bears inhabiting the Kenai Peninsula. Home range sizes varied widely in area (Fig. 2), and spatially, as bears often moved linearly within mountain valleys and through passes, thereby defying the notion of a symmetrical home range (G Harris and S Farley, unpublished data).

Unfortunately, SECR models are not yet designed for data gained by a targeted approach (JA Royle, pers. comm. 2013). If activity centers were built for bears trapped at streams, it would imply that bears only occurred there. This would severely bias the true distribution of bear's activity centers across the landscape. In reality, bear movements follow a general pattern whereby they periodically visit streams as they wander throughout the wider landscape (G Harris and S Farley, unpublished data). Therefore, by using MARK to compute abundance indexes, it ensured that each of the scenarios contained similar sampling and quantitative assumptions.

## Density estimation

An important part of abundance estimation is defining the sampling area, and determining if that area is open or closed to species movement. For targeted sampling, defining the study area requires knowing the "attraction distance" of the species to the resource concentrations (where sampling occurs). We do not know the attraction distance of brown bears to anadromous streams. However, in our study, all bears spanning the peninsula visited anadromous streams. Because the attraction distance was not exceeded for these bears, a targeted sampling design actually conducted on the Kenai Peninsula could estimate density. Obviously, this is not the case for many other species and locations. For them,

additional work will probably be required to determine the attraction distance, which would in turn define the sampling frame. Fortunately, there is precedence for such efforts, as biologists on the African continent estimate abundance of lions and hyenas based on their attraction to vocalization lures. Biologists quantified the attraction distance of these carnivores to the lure, to estimate species density (*Ogutu & Dublin, 1998*).

## CONCLUSIONS

When sampling cryptic animals inhabiting thickly vegetated and expansive areas, with DNA, camera traps or other CMR methodology, it can challenge abilities to produce accurate and affordable abundance indices. Yet targeted sampling met these criteria, by sampling where individuals gathered in smaller areas, at biologically important places and times. It generated encounter rates four times higher than grid sampling, capture probabilities 11% higher, and 60% higher recapture probabilities. This reduced capture heterogeneity (*Williams, Nichols & Conroy, 2002*; *Lukacs, 2009*). By generating a CV of 12%, targeted sampling was precise. Better precision enables measuring changes in abundances quicker, with trends more likely to reflect true changes in the population and not sampling or biological artifacts. Lastly, targeted sampling was more economical, as it used half the effort of the grid.

Since our simulation had a known number of bears, it enabled us to evaluate the veracity of the grid and targeted designs. Hence, our simulation provided insight into sampling approaches and assumptions for grid and targeted sampling that would be difficult to provide with field studies. However, conducting field studies to test and verify our findings, or refine methodologies, forms a logical next step. Fortunately, for targeted sampling, some field studies already provide insight. For instance, *Sawaya et al. (2012)* found that collecting hair from bear rubs generated higher detection rates and capture probabilities than traps in grids, for all classes of bears. *Kendall et al. (2009)* also relied on collecting hair on bear rubs and from physical captures to improve capture probabilities, as samples gained from grid traps alone provided inaccurate results.

Grids blanket entire study areas under the guise of ensuring equal probability of capture. This approach should work, provided the cells are small enough to encapsulate home ranges of all individuals. This requires home range data, which can be expensive and laborious to procure. If the average home range size is used, or were cell size based on home range data gained elsewhere, or home range sizes changed between surveys, then cell sizes may be too large. Then, users of grid sampling may assume that their techniques meet the sampling assumptions required to estimate population abundance, when in actuality, not all individuals are available for capture. Worryingly, many studies have not confirmed that the cell sizes they employ are appropriate (e.g., *Woods et al., 1999*; *Boulanger, Stenhouse & Munro, 2004*; *Mowat et al., 2005*; *Boulanger et al., 2008a*; *Boulanger et al., 2008b*; *Kendall et al., 2008*; *Sawaya et al., 2012*). Therefore, their results may represent estimates or indices, and if the latter, how the index relates to the estimate is unknown.

Criticism of the targeted approach centers on subjective or convenience sampling, largely regarded as poor practice. As with grids, to ensure that subjective sampling does not

occur, studies should demonstrate that individuals have a nonzero probability of capture (to produce estimates). Unfortunately, other studies using targeted sampling have not justified if all individuals in the sampling frame have a nonzero probability of capture, so whether these studies meet this assumption also remains unknown (*Karanth & Nichols, 1998*; *Kendall et al., 2009*; *Sawaya et al., 2012*).

We found that grid and targeted sampling of brown bears on the Kenai Peninsula had a few individuals with zero probability of capture, and therefore did not meet this assumption. The extent that serious problems in wildlife management or conservation stem from studies sampling with grids or targeted designs and presenting estimates that could represent indices remains uncertain. However, in keeping with our results, the biases are likely to be negative, thereby erring conservatively. Although, the trends for grids would likely be less informative than the trends reported from targeted designs, because the grid designs were more imprecise, which could mask population changes when they occur.

The targeted approach relied on identifying biologically important resource concentrations, and sampling at those sites. For bears, salmon runs may change in timing between periods, so bears could visit them outside the sampling period. Or, some resource concentrations may be unknown, and therefore not sampled. If so, the cost of inadequately covering all resource concentrations is larger bias. However, population indices from the targeted design maintained precision, rendering them useful for evaluating trend (provided traps were moved between sessions). Therefore, the outcome of heterogeneity in animal captures for trend analyses manifests in grid sampling losing utility as trap density declines (via larger grid cells), while trends from targeted sampling remained useful.

Biologists must weigh the theoretical robustness of sampling and modeling procedures with the logistical constraints of a given sampling design. Clearly, both grid and targeted sampling have benefits and flaws. Before employing these designs, practitioners should evaluate their sampling assumptions, the methodological drawbacks, and the utility of the results generated. The grid and targeted designs are not appropriate for all species in every situation. Neither approach is a panacea.

Targeted sampling could assist with indexing abundance of other species that concentrate at biological resources – be it trails, water holes, mineral licks, or animal latrines (*Karanth, Nichols & Kumar, 2004*). For example, travel routes and trail intersections draw tigers (*Karanth & Nichols, 1998*). The thick lowland forests of the Southwestern Amazon contain sparsely distributed macaws. Macaws gather by the hundreds to take clay from exposed riverbanks (*Diamond, 1999*). Similarly, burned patches in lowland Nepal attract axis deer (*Axis axis*) from forests (*Moe & Wegge, 1997*), while alpine ibex (*Capra ibex*) and bighorn sheep (*Ovis canadensis*) vacate rugged terrain to frequent natural and anthropogenic sources of salt and minerals (*Schmidt, Rutherford & Bodenham, 1978*; *Watts & Schemnitz, 1985*; *Bassano et al., 2003*). Walruses (*Odobenus rosmarus*) aggregate at terrestrial haul-out sites (*Lydersen, Aars & Kovacs, 2008*) and elephants congregate at water sources (*Harris et al., 2008*).

Globally, many species inhabit expansive and inaccessible areas, and have sparse distributions within them. Because these species are difficult to sample, data describing

their abundances are often deficient. Many of these animals are of high conservation concern. Biologists require methods to sample these animals effectively and economically, for indexing abundance and reporting trends, to subsequently conserve and manage their populations. In extensive areas where animals concentrate in confined locations, targeted sampling could provide such an approach.

## ACKNOWLEDGEMENTS

We thank the Alaska Department of Fish and Game for continuing to support telemetry of brown bears. Pilots J DeCreft, R Ernst, and M Litzen ensured safe captures and data collection. M Conroy, T Debruyn, J Morton and J Sanderson were generous with time, knowledge and ideas. A Jacobson, DH Johnson and other anonymous reviewers provided helpful comments to strengthen this manuscript. The findings and conclusions in this article are those of the author(s) and do not necessarily represent the views of the U.S. Fish and Wildlife Service. The use of trade, firm, or product names is for descriptive purposes only and does not imply endorsement by the U.S. Government.

### Funding

The Alaska Department of Fish and Game, the USDA Forest Service, National Park Service and U.S. Fish and Wildlife Service provided logistical and financial backing. The funders had no role in study design, data collection and analysis, decision to publish, or preparation of the manuscript.

### Competing Interests

The authors declare that there are no competing interests.

### Author Contributions

- Grant Harris conceived and designed the experiments, performed the experiments, analyzed the data, wrote the paper.
- Sean Farley and Jeff Selinger conceived and designed the experiments, contributed reagents/materials/analysis tools, wrote the paper.
- Gareth J. Russell and Matthew J. Butler conceived and designed the experiments, analyzed the data, wrote the paper.

### Animal Ethics

The following information was supplied relating to ethical approvals (i.e., approving body and any reference numbers):

All field and capture methods were approved by Alaska Department of Fish and Game, Animal Care and Use Committee, Assurance No. 06-03.

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
