# Peer review of "Sampling designs matching species biology produce accurate and affordable abundance indices"

_PeerJ, doi:10.7717/peerj.227_

## Round 0.1 · original submission · Minor Revisions

· Academic Editor

Minor Revisions

As you can see, the two reviewers like your work and have recommended you make only minor changes to it. The instructions are clear — make the changes and let us see how you've done so.

Reviewer 1 ·

Basic reporting

No Comments

Experimental design

No Comments

Validity of the findings

No Comments

Additional comments

My main comment on the paper deals with the issue of density estimation. Often, when managers or researchers employ grid designs, their primary objective is to estimate density of animals, which requires an estimate of abundance. I think it would be worthwhile for the authors to discuss the limitations of the targeted design when density estimation is the primary objective. For example, when bears are concentrated near the salmon streams, the density in those areas is inflated for the period of time that salmon are available, then the bears disperse. Given that many studies will not have radio collared animals, the maximum attraction distance from the concentrated resource are largely going to be unknown. So this method would probably not work for studies that have the goal of estimating density rather than just simply abundance.

Lines 228-230. I would disagree with the contention that shortening the sampling session automatically confers a time or cost benefit. I think this would only be for smaller study areas where researchers could reasonably be expected to check all the traps within the shortened session without having to hire extra personnel to do so.
For large study areas with a given number of traps or grid cells, having longer sessions, thus allowing the limited number of field personnel that can be hired on a fixed budget to check all traps within each session, redeploy them, then start over. For large projects, with lots of traps a shorter capture session would necessitate that you have more personnel available to check all the traps within each sampling session.

Minor comments
Line 39. It would be informative to provide a few of these species as examples parenthetically.

Line 97. change abundances to abundance estimates or estimates of abundance.

Lines 106-107, 110-112, 122-124. Some of this paragraph and the next is really more appropriate for the methods section rather than the end of the introduction.

Lines 131-133 These two sentences could be combined with the above paragraph.

Lines 181-183. Didn't you just warn against this practice in the introduction? If so, why employ it here?

Lines 223-226, 232-234. These are actually results, not methods.

Lines 250-251. Why not include all parameter estimates from all models in the model averaging (i.e., model average across all models)?

Line 284. What was the average proportion of the fixes for each bear that were considered "captures"?

Line 285. vague-- how much lower?

Line 306-309. Awkward sentence structure.

Line 444. Depending on the management objectives for the population (i.e., increase, decrease, keep stable), bias in either direction can be problematic.

·

Basic reporting

The MS is well-written with little jargon and short sentences.

Experimental design

My one primary concern with the MS is the lack of clarity in Methods on how the simulations were done. It is unclear what aspects of the study were done in practice, on the peninsula, or in a simulation. Were the GPS tracks of individual animals input into simulations in which the camera trap placement was modified? Greater distinction is needed here.
Line467 starts with suggesting how difficult it is to estimate distances which lures attract bears; but this is done when estimating population size with call-up stations in Africa for lions, hyenas.

Validity of the findings

No comments

Additional comments

There are a few minor spelling mistakes in MS and text for Figures/Tables. Few other comments: is there a source and example for home ranges having significant variation within a population (Line55); introduce applicability of methods not just to resource concentrations but animal trails/rubs etc earlier (Line84); paragraph starting at Line106 seems misplaced - consider placing in methods; introduction of studies to capture DNA in Line149 is totally from left field, place in better context; is first assumption that bears range widely immed after leaving dens a reasonable assumption? - provide source; consider explicitly mentioning how many cameras you used (P starting with line 199); Line244 its 4th not 3rd model; Line 252 needs new heading Evaluation Measures?; Line302 cell size should be changed to edge lengths of 7,9,11?; Line576 you state you met the assumption but it seems you did not - there were bears with prob of capture at zero in both studies;

---

## Round 0.2 · accepted · Accept

· Academic Editor

Accept

Thank you for quickly and effectively responding to the many minor suggestions the reviewers made on your earlier draft. I think you have addressed all these comments satisfactorily.